# Role of intraoperative oliguria in risk stratification for postoperative acute kidney injury in patients undergoing colorectal surgery with an enhanced recovery protocol: A propensity score matching analysis

**Jung-Woo Shim[1], Kyoung Rim Kim[2], Yoonju Jung[3], Jaesik Park[1], Hyung Mook Lee[1], Yong-Suk Kim[1], Young Eun Moon[1], Sang Hyun Hong[1], Min Suk Chae**[1] *

1 Department of Anesthesiology and Pain Medicine, Seoul St. Mary's Hospital, College of Medicine, The Catholic University of Korea, Seoul, Republic of Korea, 2 Department of Anesthesiology and Pain Medicine, Bucheon St. Mary's Hospital, College of Medicine, The Catholic University of Korea, Seoul, Republic of Korea, 3 Department of Surgery, Seoul St. Mary's Hospital, College of Medicine, The Catholic University of Korea, Seoul, Republic of Korea

* shscms@gmail.com

## Abstract

### Background

The enhanced recovery after surgery (ERAS) protocol for colorectal cancer resection recommends balanced perioperative fluid therapy. According to recent guidelines, zero-balance fluid therapy is recommended in low-risk patients, and immediate correction of low urine output during surgery is discouraged. However, several reports have indicated an association of intraoperative oliguria with postoperative acute kidney injury (AKI). We investigated the impact of intraoperative oliguria in the colorectal ERAS setting on the incidence of postoperative AKI.

### Patients and methods

From January 2017 to August 2019, a total of 453 patients underwent laparoscopic colorectal cancer resection with the ERAS protocol. Among them, 125 patients met the criteria for oliguria and were propensity score (PS) matched to 328 patients without intraoperative oliguria. After PS matching had been performed, 125 patients from each group were selected and the incidences of AKI were compared between the two groups. Postoperative kidney function and surgical outcomes were also evaluated.

### Results

The incidence of AKI was significantly higher in the intraoperative oliguria group than in the non-intraoperative oliguria group (26.4% *vs.* 11.2%, respectively, $P = 0.002$). Also, the eGFR reduction on postoperative day 0 was significantly greater in the intraoperative oliguria than non-intraoperative oliguria group (−9.02 *vs.* −1.24 mL/min/1.73 m$^2$ respectively,

**Data Availability Statement:** All relevant data are within the paper and its Supporting Information files.

**Funding:** The author(s) received no specific funding for this work.

**Competing interests:** The authors have declared that no competing interests exist.

**Abbreviations:** ERAS, enhanced recovery after surgery; AKI, acute kidney injury; ASA classification, American Sociey of Anesthesiologists Physical Status Classification; MBP, mean blood pressure; PCA, patient-controlled analgesia; PS, propensity score; BMI, body mass index; TNM stage, TNM Classification of Malignant Tumors stage; KDIGO, Kidney Disease Improving Global Outcomes; eGFR, estimated glomerular filtration rate; POD, postoperative day.

$P < 0.001$). In addition, the surgical complication rate was higher in the intraoperative oliguria group than in the non-intraoperative oliguria group (18.4% *vs.* 9.6%, respectively, $P = 0.045$).

## Conclusions

Despite the proven benefits of perioperative care with the ERAS protocol, caution is required in patients with intraoperative oliguria to prevent postoperative AKI. Further studies regarding appropriate management of intraoperative oliguria in association with long-term prognosis are needed in the colorectal ERAS setting.

## Introduction

Balanced perioperative fluid therapy is a key component of enhanced recovery after surgery (ERAS) protocols [1]. According to recent guidelines from the ERAS Society, zero-balance and goal-directed fluid therapies are both appropriate for intraoperative fluid administration. However, zero-balance fluid therapy is especially recommended for low-risk patients to avoid fluid overload. Zero-balance fluid therapy refers to a regimen involving restrictive fluid administration and provision of minimal maintenance fluid, while ignoring third space loss or volume deficit that results from preoperative fasting [2–4]. In contrast, guidelines regarding the management of surgical patients and sepsis recommend guiding fluid administration by monitoring the urine output [5, 6]. In addition, oliguria, which is traditionally defined as urine output < 0.5 mL/kg/h, has been regarded as an early marker of renal hypoperfusion and acute kidney injury (AKI). However, in a randomized trial, Puckett et al. [7] concluded that targeting a lower urine output allowed administration of a smaller amount of intraoperative fluid without significant effects on clinical outcomes. Furthermore, the association of perioperative fluid overload with worsening of postoperative morbidity is well-established [8–12].

In an ERAS setting, with zero-balance fluid therapy, perioperative oliguria alone is often not considered an important indicator for hypoperfusion [1, 13]. Many factors other than renal perfusion influence urine output, including overall hemodynamic status, sympathetic activity, and the effects of hormones, such as aldosterone and antidiuretic hormone. These represent other reasons to advocate avoidance of additional fluid administration to promptly correct perioperative oliguria. However, several recent studies demonstrated that intraoperative oliguria was associated with postoperative AKI in patients undergoing major abdominal surgery. Mizoda et al. [14] and Myles et al. [15] reported that intraoperative oliguria was significantly associated with increased incidence of postoperative AKI. Shive et al. [16] reported that oliguria for ≥ 120 minutes was independently associated with the occurrence of postoperative AKI. These studies suggested that interpretation of intraoperative oliguria was important for prevention of postoperative AKI. In addition, caution is reportedly required with regard to increased AKI after implementation of colorectal ERAS protocols [17, 18]. However, the prior studies did not evaluate the impact of intraoperative oliguria on postoperative AKI. This study was performed to investigate the association of intraoperative oliguria with postoperative AKI in patients undergoing laparoscopic colorectal cancer resection with an ERAS protocol.

## Patients and methods

### Ethical considerations

The Institutional Review Board and Ethics Committee of Seoul St. Mary's Hospital, a tertiary academic teaching hospital, approved this study protocol (approval number: KC19RESI0598); the study was performed in accordance with the Declaration of Helsinki. The requirement for informed consent was waived due to the retrospective nature of the study.

### Study population in ERAS protocol

As shown in S1 Table, our hospital launched the ERAS protocol for perioperative care of patients undergoing colorectal cancer resection after a 3-month trial period. The components of the ERAS program were selected with consideration of several ERAS-related principles [13, 19, 20]. The protocol was registered in January 2017, and the data were recorded for patients who underwent treatment with the ERAS protocol [21]. At the preoperative visit, patients were identified as candidates for the ERAS program. The inclusion criteria for the colorectal ERAS program were as follows: (1) colorectal adenocarcinoma; (2) elective surgery; and (3) voluntary consent for the ERAS protocol. Exclusion criteria were as follows: (1) severe illness affecting surgery or general anesthesia; (2) emergency operation; and/or (3) infectious state.

Colorectal cancer resections were performed in 565 patients from January 2017 to August 2019. Among them, 32 patients with American Society of Anesthesiologists Physical Status Classification (ASA classification) $\geq$ 3, 21 patients who underwent emergency operations, 51 patients who refused or were unable to understand the ERAS protocol were excluded from the colorectal ERAS program. In addition, eight patients with primary open or conversion surgeries were excluded from the study. In total, this study retrospectively investigated 453 patients who underwent laparoscopic colorectal cancer surgery with the ERAS program.

### Surgery

Laparoscopy-based surgeries were performed by experienced surgeons in patients with the ERAS protocol. Operative procedures included hemicolectomy, extended hemicolectomy, segmental colectomy, anterior resection, abdominoperineal resection, subtotal colectomy, total colectomy, total proctocolectomy, and intersphincteric resection with coloanal anastomosis.

### General anesthetic management

General anesthesia was induced with propofol (2 mg/kg) and rocuronium (0.8 – 1 mg/kg), followed by tracheal intubation; multiple monitoring modalities were used, including electrocardiography, pulse oximetry, non-invasive blood pressure measurement, and bispectral index assessment. In addition, end-tidal carbon dioxide level and esophageal temperature were routinely monitored. A radial arterial line was cannulated to monitor arterial blood pressure or dynamic indices using arterial blood pressure data. Anesthesia was maintained using inhaled desflurane with medical air/oxygen, with bispectral index < 40. Continuous rocuronium infusion at a dose of 0.3 – 0.6 mg/kg/h was administered to maintain sufficient neuromuscular relaxation until peritoneal closure. Hypotension, defined as mean blood pressure (MBP) reduction < 60 – 65 mmHg, was initially treated with 5 mg of ephedrine, then with 50 µg of phenylephrine. If hypotension persisted without severe blood loss, continuous norepinephrine infusion was initiated at a dose of 0.03 – 0.10 µg/kg/min. At the end of surgery, neuromuscular blockade was reversed with sugammadex (4 mg/kg) under ventilation with 100% $O_2$. The endotracheal tube was removed from the trachea after confirmation of spontaneous eye opening and train-of-four ratio $\geq$ 90%.

Preoperative epidural analgesia was performed at the discretion of the attending surgeons and anesthesiologists, and epidural patient-controlled analgesia (PCA) was continuously infused postoperatively. In patients without epidural analgesia, fentanyl-based intravenous PCA was infused for postoperative analgesia. If patients experienced postoperative pain > 4 on a visual analog scale despite the use of epidural or intravenous PCA, the following intravenous rescue analgesics were provided: first ketorolac (30 mg), then pethidine (25 μg).

## Intraoperative fluid therapy

Restrictive fluid therapy was used for intraoperative fluid administration, defined as maintenance fluid with 1 – 3 mL/kg/h and additional fluid bolus for blood loss. Balanced salt solutions were used for maintenance of perioperative electrolyte balance [22, 23]. The ratios of fluid administration for blood loss were 1.5:1 with crystalloid and 1:1 with colloid [24]. Intraoperative fluid therapy was guided using a FloTrac system (Edwards Lifesciences, Irvine, CA, USA) or CardioQ–Oesophageal Doppler monitoring (Deltex, Chichester, UK), in accordance with the preference of the attending anesthesiologist. Based on the findings in several previous reports [13, 25], the reduced hourly urine output itself was not managed with specific intraoperative interventions, such as fluid resuscitation therapy or injection of diuretics. Instead, oliguric patients were observed unless clinical signs of hypovolemia occurred. No patient in this study received diuretics during surgery.

## Definition of oliguria during surgery

For all patients, the average intraoperative urine was calculated by dividing the total urine output by the length of surgery and the measured body weight. Intraoperative oliguria was defined as average urine output < 0.5 mL/kg/h during surgery. Review of intraoperative data for 453 patients in the ERAS program revealed that 125 patients showed intraoperative oliguria and were classified as the intraoperative oliguria group. The remaining 328 patients were classified as the non-intraoperative oliguria group.

## Propensity score matching variables

The 125 patients in the intraoperative oliguria group were propensity score (PS) matched to 328 patients in the non-intraoperative oliguria group. Patients were matched according to age, sex, body mass index (BMI), ASA classification, preoperative hemoglobin level, preoperative albumin level, preoperative creatinine level, preoperative MBP, preoperative heart rate, length of surgery, segmental resection, epidural analgesia, intraoperative norepinephrine dose, intraoperative phenylephrine dose, average intraoperative MBP, average intraoperative heart rate, intraoperative fluid balance, and postoperative TNM Classification of Malignant Tumors stage (TNM stage). The intraoperative doses of norepinephrine and phenylephrine were calculated by dividing the total doses by the patient's body weight. Average intraoperative MBP and heart rate were calculated by dividing the sum of MBP and heart rate, respectively, at the start of anesthesia, at 1/3 of the duration of the surgery, at 2/3 of the duration of the surgery, and at the end of anesthesia, by 4 in each patient. After PS matching had been performed, 125 patients from each group were selected.

## Definition of AKI

Both the incidence and extent of AKI, as classified by the Kidney Disease Improving Global Outcomes (KDIGO) staging system, were recorded [26] (S2 Table). Postoperative kidney function was quantified based on the estimated glomerular filtration rate (eGFR), calculated using

the Modification of Diet in Renal Disease formula: eGFR = 175 × standardized serum creatinine$^{-1.154}$ × age$^{-0.203}$ × 1.212 (if black) × 0.742 (if female) [27].

## Clinical variables

All data were collected from the digital medical records of Seoul St. Mary's Hospital. Demographic information was recorded in all patients; this information included age, sex, BMI, ASA classification, underlying disease, smoking history, and bowel preparation. In addition, preoperative laboratory test results (within 30 days of surgery) were collected for each patient; these included levels of serum hemoglobin, albumin, and creatinine. Preoperative vital signs were recorded in all patients after they drank a can of complex carbohydrate solution, 2 h before operation. Intraoperative characteristics included case length, procedure type, approach type (i.e., open, laparoscopic, or robot-assisted laparoscopic), ostomy creation, epidural block, intraoperative vasoconstrictor use (including the total doses of norepinephrine and phenylephrine), and fluid balance (i.e., crystalloid infusion, colloid infusion, urine output, and estimated blood loss). In addition, intraoperative vital signs (e.g., blood pressure, heart rate, and body temperature) were recorded at 5-minute intervals during surgery. Daily measurements of creatinine level and postoperative fluid balance (e.g., amounts of intravenous and oral fluids, urine output, and blood loss) were also recorded. Postoperative outcomes included any morbidity (i.e., ileus, leakage, and others), reoperation, and mortality. TNM stage was based on postoperative pathological findings. Postoperative morbidity, reoperation rate, and mortality rate were reviewed until 1 month after surgery. Any postoperative complications other than renal problems were recorded as general morbidity. The following surgical complications were recorded: ileus, leakage, surgical site infection, postoperative bleeding, wound dehiscence, chylous ascites and intra-abdominal spillage.

## Statistical analysis

The normality of continuous data was assessed using the Shapiro–Wilk test. Descriptive statistics for categorical variables were reported as frequency (%) and continuous variables as mean (standard deviation) or median (interquartile range). PS matching analysis was performed to reduce the impacts of potential confounding factors on intergroup differences, based on intraoperative oliguria. PSs were derived to match patients at a 1:1 ratio using greedy matching algorithms without replacement. The $\chi^2$ test or Fisher's exact test was used to compare categorical variables, as appropriate. Student's $t$ test or the Mann–Whitney U test was used to compare continuous variables, as appropriate. The association of intraoperative oliguria with postoperative AKI was evaluated by multivariable logistic regression analysis with PS adjustment. The values are presented as odds ratios with 95% confidence intervals. All statistical tests were two-sided, and differences with $P < 0.05$ were considered statistically significant. Analysis was performed using R software version 2.10.1 (R Foundation for Statistical Computing, Vienna, Austria) and SPSS software version 24.0 (IBM Corp., Armonk, NY, USA).

## Results

### Demographic characteristics of the entire study population

We included 236 male (52.1%) and 217 female patients (47.9%). The average ± standard deviation age and BMI were 64 ± 12 years and 23.8 ± 3.4 kg/m$^2$, respectively. Of all patients, 139 (30.7%) were of ASA classification 1 and 314 (69.3%) were of ASA classification 2. The average ± standard deviation preoperative hemoglobin, albumin, and creatinine levels were 12.9 ± 1.9 g/dL, 4.2 ± 0.5 g/dL, and 0.9 ± 0.3 mg/dL, respectively. The average ± standard

deviation blood pressure and heart rate were 94 ± 15 mmHg and 72 ± 10 /min, respectively. No patient exhibited preoperative hypotension (MBP < 65 mmHg) or tachycardia (heart rate > 100/min).

## Comparison of clinical characteristics before and after PS matching

Comparisons of preoperative, intraoperative, and pathological characteristics between the PS-matched intraoperative oliguria and non-intraoperative oliguria groups are shown in Table 1. There were no significant differences between the two groups in terms of the variables used for PS matching (i.e., age, sex, BMI, ASA classification, preoperative hemoglobin level, preoperative albumin level, preoperative creatinine level, preoperative MBP, preoperative heart rate, length of surgery, segmental resection status, epidural analgesia status, intraoperative norepinephrine dose, intraoperative phenylephrine dose, average intraoperative MBP, average intraoperative heart rate, average intraoperative fluid balance, and TNM stage).

## Comparison of intraoperative fluid balance before and after PS- matching

Comparisons of fluid input, urine output, and hemorrhage between the intraoperative oliguria and non-intraoperative oliguria groups are shown in Table 2. After PS matching had been performed, patients in the intraoperative oliguria group showed significantly less crystalloid infusion (600 *vs*. 800 mL, respectively, *P* = 0.016), total intravenous fluid (3.32 *vs*. 4.48 mL/kg/h, respectively, *P* = 0.001), and urine output (0.33 *vs*. 0.96 mL/kg/h, respectively, *P* < 0.001), compared to patients in the non-intraoperative oliguria group. Also, the estimated blood loss was lower in the intraoperative group compared to the non-intraoperative oliguria group (50 mL *vs*. 50 mL, respectively, *P* = 0.033).

## Postoperative kidney function

Postoperative AKI occurred in 17.4% of the overall study population. The incidence of AKI was significantly higher in the intraoperative oliguria group than in the non-intraoperative oliguria group, as shown in Fig 1 (26.4% *vs*. 11.2%, respectively, *P* = 0.002). The majority of patients with AKI exhibited KDIGO stage 1 disease (21.6% *vs*. 9.6%, respectively, *P* = 0.009) and the incidence of KDIGO stage 2 was not significantly different between the two groups (4.0% *vs*. 0.8%, respectively, *P* = 0.213); one patient in each group was diagnosed with KDIGO stage 3 disease. Perioperative eGFR values are shown in Table 3. There were no significant differences in preoperative or postoperative eGFR values between the two groups. However, the degree of reduction in eGFR on postoperative day (POD) 0 was significantly greater in the intraoperative oliguria group than in the non-intraoperative oliguria group (−9.0 *vs*. −1.2 mL/min/1.73 m², respectively, *P* < 0.001).

## Daily fluid balance

Daily fluid intake and urine output data for the two PS-matched groups are shown in Fig 2. Urine output on POD 0 was significantly lower in the intraoperative oliguria group than in the non-intraoperative oliguria group (0.47 *vs*. 0.72 mL/kg/h, respectively, *P* < 0.001). Although the oral and total fluid inputs on PODs 0–2 did not differ significantly between the groups, the amount of intravenous fluid required by the intraoperative oliguria group was significantly higher on POD 2 than that required by the non-intraoperative oliguria group (705 *vs*. 589 mL, respectively, *P* = 0.031).

**Table 1. Comparisons of preoperative intraoperative, and pathological factors between the non-intraoperative oliguria and intraoperative oliguria groups before and after propensity score matching analysis.**

| Group | Before PS matching | | | | After PS matching | | | |
|---|---|---|---|---|---|---|---|---|
| | Non-intraoperative oliguria (*n* = 328) | Intraoperative oliguria (*n* = 125) | *P*-value | SD | Non-intraoperative oliguria (*n* = 125) | Intraoperative oliguria (*n* = 125) | *P*-value | SD |
| Preoperative factors | | | | | | | | |
| Age (years) | 64 ± 12 | 64 ± 12 | 0.574 | −0.059 | 64 ± 11 | 64 ± 12 | 0.736 | −0.040 |
| Male sex | 155 (47.3%) | 81 (64.8%) | 0.001 | −0.366 | 76 (60.8%) | 81 (64.8%) | 0.513 | −0.083 |
| BMI > 25 kg/m$^2$ | 88 (26.8%) | 57 (45.6%) | <0.001 | 0.375 | 57 (45.6%) | 57 (45.6%) | 1.000 | 0.000 |
| ASA classification | | | 0.935 | 0.009 | | | 1.000 | 0.000 |
| ASA 1 | 101 (30.8%) | 38 (30.4%) | | | 38 (30.4%) | 38 (30.4%) | | |
| ASA 2 | 227 (69.2%) | 87 (69.6%) | | | 87 (69.6%) | 87 (69.6%) | | |
| Preoperative hemoglobin (g/dL) | 12.8 ± 1.8 | 13.1 ± 2.2 | 0.100 | 0.166 | 13.0 ± 1.9 | 13.1 ± 2.2 | 0.635 | 0.056 |
| Preoperative albumin (g/dL) | 4.1 ± 0.5 | 4.2 ± 0.4 | 0.034 | 0.234 | 4.2 ± 0.4 | 4.2 ± 0.4 | 0.988 | 0.002 |
| Preoperative creatinine (mg/dL) | 0.8 ± 0.2 | 0.90 ± 0.31 | 0.013 | 0.222 | 0.9 ± 0.3 | 0.9 ± 0.3 | 0.890 | 0.016 |
| Preoperative mean blood pressure (mmHg) | 95 ± 15 | 94 ± 14 | 0.481 | −0.078 | 94 ± 16 | 94 ± 14 | 0.696 | −0.052 |
| Preoperative heart rate (/min) | 72 ± 11 | 72 ± 10 | 0.864 | −0.019 | 72 ± 11 | 72 ± 10 | 0.780 | 0.038 |
| Intraoperative factors | | | | | | | | |
| Case length (min) | 177 ± 66 | 192 ± 67 | 0.040 | 0.214 | 189 ± 65 | 192 ± 67 | 0.722 | 0.044 |
| Segmental resection | 90 (27.4%) | 26 (20.8%) | 0.148 | 0.163 | 31 (24.8%) | 26 (20.8%) | 0.451 | 0.098 |
| Epidural block | 156 (48.2%) | 61 (48.8%) | 0.905 | 0.013 | 62 (49.6%) | 61 (48.8%) | 0.899 | −0.016 |
| Norepinephrine dose (μg/kg) | 0.00 (0.00–0.00) | 0.00 (0.00–0.00) | 0.088 | 0.193 | 0.00 (0.00–0.00) | 0.00 (0.00–0.00) | 0.424 | 0.078 |
| Phenylephrine dose (μg/kg) | 0.00 (0.00–0.00) | 0.00 (0.00–1.41) | 0.062 | 0.166 | 0.00 (0.00–1.03) | 0.00 (0.00–1.41) | 0.833 | −0.013 |
| Average mean blood pressure (mmHg) | 88 ± 9 | 88 ± 9 | 0.730 | 0.036 | 88 ± 9 | 88 ± 9 | 0.694 | −0.051 |
| Average heart rate (/min) | 73 ± 13 | 73 ± 9 | 0.837 | −0.027 | 72 ± 9 | 73 ± 9 | 0.792 | 0.033 |
| † Fluid balance (mL/kg/h) | 2.51 (1.28–4.37) | 2.69 (1.64–4.44) | 0.260 | 0.117 | 2.70 (1.44–4.53) | 2.69 (1.64–4.44) | 0.877 | 0.067 |
| Pathological factors | | | | | | | | |
| TNM stage | | | 0.224 | −0.108 | | | 0.422 | −0.059 |
| 0 | 11 (3.4%) | 1 (0.8%) | | | 5 (4.0%) | 1 (0.8%) | | |
| 1 | 96 (29.3%) | 46 (36.8%) | | | 38 (30.4%) | 46 (36.8%) | | |
| 2 | 84 (25.6%) | 31 (24.8%) | | | 30 (24.0%) | 31 (24.8%) | | |
| 3 | 112 (34.1%) | 42 (33.6%) | | | 45 (36.0%) | 42 (33.6%) | | |
| 4 | 25 (7.6%) | 5 (4.0%) | | | 7 (5.6%) | 5 (4.0%) | | |

Data are presented as the mean ± standard deviation, median with interquartile range in parentheses, or as number with percentage in parentheses.

† Fluid balance was defined as the difference between total fluid input and output during surgery.

**Abbreviations:** PS, propensity score; SD, standardized difference; BMI, body mass index; ASA classification, American Society of Anesthesiologists Physical Status Classification; TNM stage, TNM Classification of Malignant Tumors stage.

## Major outcomes

Major postoperative outcomes in the two PS-matched groups are shown in Table 4. There were no significant differences in terms of the postoperative length of stay, general morbidity, or Clavien-Dindo classification between the two groups. However, the rate of surgical

**Table 2. Comparisons of fluid input, urine output, and hemorrhage between non-intraoperative oliguria and intraoperative oliguria groups before and after propensity score matching analysis.**

| Group | Before PS matching | | | After PS matching | | |
|---|---|---|---|---|---|---|
| | Non-intraoperative oliguria (*n* = 328) | Intraoperative oliguria (*n* = 125) | *P*-value | Non-intraoperative oliguria (*n* = 125) | Intraoperative oliguria (*n* = 125) | *P*-value |
| Crystalloid infusion (mL) | 610 (400 – 1000) | 600 (400 – 1025) | 0.968 | 800 (500–1100) | 600 (400 – 1025) | 0.016 |
| Colloid infusion (mL) | 0 (0 – 0) | 0 (0 – 0) | 0.438 | 0 (0–0) | 0 (0 – 0) | 0.287 |
| Total IV fluid (mL/kg/h) | 4.17 (2.81 – 6.02) | 3.32 (2.07 – 5.30) | 0.001 | 4.48 (2.95–6.17) | 3.32 (2.07 – 5.30) | 0.001 |
| Urine output (mL/kg/h) | 1.00 (0.79 – 1.47) | 0.33 (0.24 – 0.39) | < 0.001 | 0.96 (0.72–1.45) | 0.33 (0.24 – 0.39) | < 0.001 |
| EBL (mL) | 50 (50 – 100) | 50 (30 – 100) | 0.138 | 50 (50–100) | 50 (30 – 100) | 0.033 |

Data are presented as the median with interquartile range in parentheses.

**Abbreviations:** IV, intravenous; EBL, estimated blood loss

complications was significantly higher in the intraoperative oliguria group than in the non-intraoperative oliguria group (18.4% *vs*. 9.6%, respectively, *P* = 0.045).

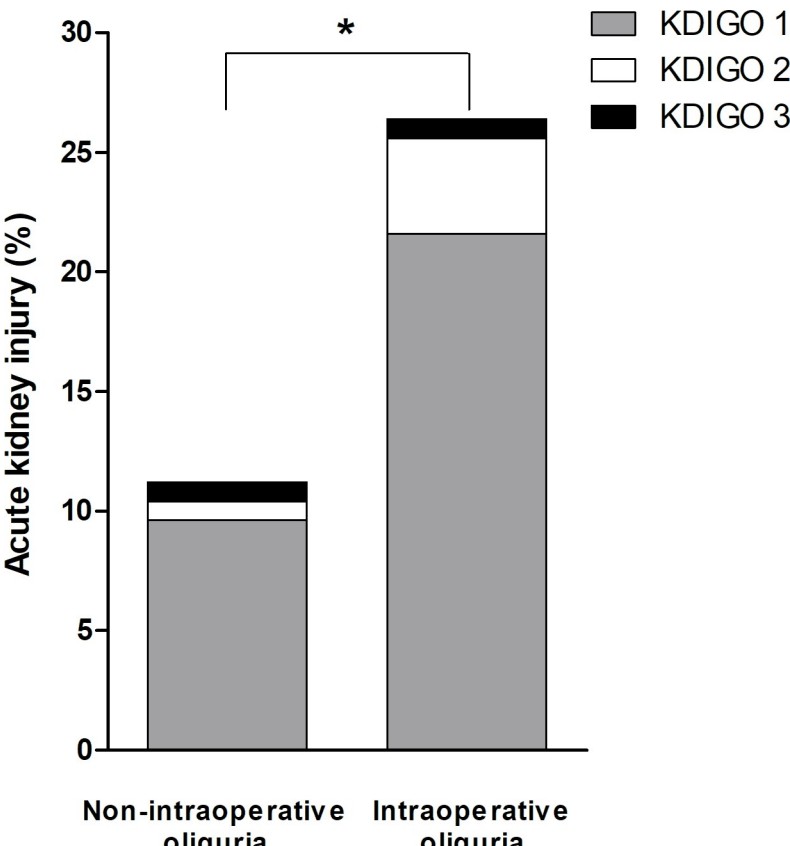

**Fig 1. Comparisons of postoperative AKI incidence and stage between non-intraoperative oliguria and intraoperative oliguria groups in propensity score-matched patients.** The incidence and stage of acute kidney injury in the two groups are shown according to the Kidney Disease Improving Global Outcomes (KDIGO) classification system. * *P* < 0.05.

**Table 3. Comparisons of perioperative eGFR values and degree of eGFR changes between non-intraoperative oliguria and intraoperative oliguria groups among propensity score-matched patients.**

| Group | Non-intraoperative oliguria (n = 125) | Intraoperative oliguria (n = 125) | P-value |
|---|---|---|---|
| **eGFR (mL/min/1.73 m²)** | | | |
| Preoperative day | 83.0 (72.5–92.8) | 86.6 (71.5 – 99.2) | 0.183 |
| POD #0 | 80.7 (71.4–89.2) | 77.4 (64.0 – 90.4) | 0.114 |
| POD #1 | 80.6 (67.0–90.4) | 77.9 (66.1 – 92.5) | 0.768 |
| POD #2 | 88.2 (71.6–99.5) | 87.5 (68.6 – 101.9) | 0.990 |
| Discharge day | 93.9 (75.9–106.9) | 90.2 (74.0 – 112.7) | 0.966 |
| One month after discharge | 83.1 (72.2–94.6) | 85.1 (73.8 – 95.6) | 0.591 |
| ‡ **eGFR change (mL/min/1.73 m²)** | | | |
| POD #0 | −1.2 (−8.6–8.5) | −9.0 (−17.1 – 1.8) | < 0.001 |
| POD #1 | −3.2 (−13.9–5.3) | −4.8 (−16.3 – 3.1) | 0.149 |
| POD #2 | 4.2 (−6.9–14.9) | 1.9 (−8.7 – 13.4) | 0.172 |
| Discharge day | 11.5 (−1.5–23.2) | 9.5 (−3.9 – 21.2) | 0.309 |
| One month after discharge | −1.3 (−8.7–9.9) | −2.3 (−12.1 – 11.0) | 0.698 |

Data are presented as the median with interquartile range in parentheses.

‡ Degree of postoperative eGFR changes, compared to preoperative baseline values.

**Abbreviations:** eGFR, effective glomerular filtration rate, calculated using the Modification of Diet in Renal Disease formula; POD, postoperative day.

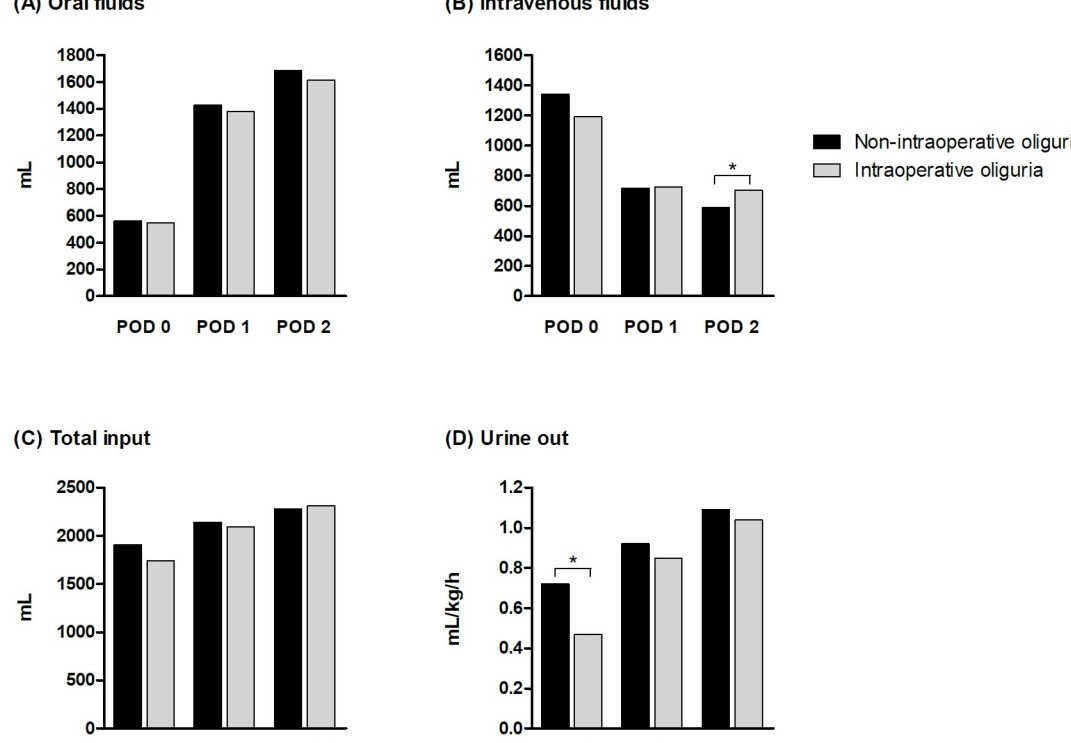

**Fig 2. Comparisons of daily fluid balance between non-intraoperative oliguria and intraoperative oliguria groups among propensity score-matched patients.** Daily fluid balance was stratified into (A) oral fluids, (B) intravenous fluids, (C) total input, and (D) urine output through postoperative day (POD) 0 – 2. Total input refers to the sum of oral fluids and intravenous fluids on each day. * P < 0.05.

**Table 4. Comparisons of postoperative outcomes between non-intraoperative oliguria and intraoperative oliguria groups among propensity score-matched patients.**

| Group | Non-intraoperative oliguria (*n* = 125) | Intraoperative oliguria (*n* = 125) | *P*-value |
|---|---|---|---|
| Postoperative length of stay (days) | 4 (4–5) | 4 (4 – 5) | 0.945 |
| General morbidity | 25 (20.0%) | 36 (28.8%) | 0.105 |
| Postoperative ileus | 6 (4.8%) | 13 (10.4%) | 0.095 |
| Anastomotic leak | 1 (0.8%) | 4 (3.2%) | 0.370 |
| Surgical complication | 12 (9.6%) | 23 (18.4%) | 0.045 |
| 30-Day reoperation | 1 (0.8%) | 2 (1.6%) | 1.000 |
| Clavien-Dindo classification | | | 0.175 |
| 1 | 101 (80.8%) | 89 (71.2%) | |
| 2 | 22 (17.6%) | 33 (26.4%) | |
| 3 | 1 (0.8%) | 3 (2.4%) | |
| 4 | 1 (0.8%) | 0 (0.0%) | |
| 5 | 0 (0.0%) | 0 (0.0%) | |

Data are presented as the median with interquartile range in parentheses, or as number with percentage in parentheses.

## Association of intraoperative oliguria with AKI within our ERAS protocol

Intraoperative oliguria was associated with AKI in the overall study population and in PS-matched patients (Table 5). After PS adjustment had been performed, intraoperative oliguria remained an independent risk factor for AKI.

## Discussion

Recent guidelines for colorectal surgery with the ERAS program state that patients with urine output < 0.5 mL/kg/h in the perioperative phase could tolerate the procedure; moreover, allowance of low urine output was associated with significantly reduced administration of intravenous fluid [1]. Because excessive fluid administration is reportedly associated with AKI, oliguria can be simply observed unless there are clear signs of dehydration or hypovolemia. However, this study showed that the incidence of AKI was significantly increased in the intraoperative oliguria group, compared to the non-intraoperative oliguria group.

Some studies showed that additional intravenous fluids or diuretics did not protect against AKI in oliguric patients [28–30]. However, there have been several studies regarding the association of intraoperative oliguria with AKI in patients undergoing abdominal surgery. Mizoda et al. [14] suggested that intraoperative oliguria, defined as urine output < 0.3 mL/kg/h, was significantly associated with increased risk of postoperative AKI in patients undergoing major abdominal surgery, whereas intraoperative urine output 0.3 – 0.5 mL/kg/h was not. In

**Table 5. Association of intraoperative oliguria with postoperative AKI.**

| Multivariable Logistic Regression Analysis | | | | |
|---|---|---|---|---|
| | *β* | Odds ratio | 95% CI | *P*-value |
| Overall patient population (*n* = 453) | | | | |
| Intraoperative oliguria adjusted for PS | 0.788 | 2.199 | 1.327 – 3.645 | 0.002 |
| PS-matched patients (*n* = 250) | | | | |
| Intraoperative oliguria adjusted for PS | 0.996 | 2.708 | 1.354 – 5.418 | 0.005 |

**Abbreviations:** CI, confidence interval; PS, propensity score.

contrast, Hur et al. [31] suggested that an optimal cutoff of mean urine output value associated with AKI after radical nephrectomy was < 1.0 mL/kg/h. The cutoff value of intraoperative urine output associated with postoperative AKI varied between these studies, because different fluid strategies were chosen in the context of various surgical settings. In addition, Myles et al. [15] reported that intraoperative oliguria, defined as urine output < 0.5 mL/kg/h, was associated with increased risk of postoperative AKI in patients undergoing major abdominal surgery.

Although the intraoperative norepinephrine and phenylephrine doses, average MBP, average heart rate, and fluid balance did not differ between the two PS-matched groups, the perioperative crystalloid infusion and total intravenous fluid requirements were significantly lower in the intraoperative than the non-intraoperative oliguria group. However, there was no significant between-group difference in the fluid intake on POD 0. It may be that the lower intraoperative fluid intake in the intraoperative oliguria group increased the risk of postoperative AKI.

The recent ERAS guidelines state that carbohydrate-containing drinks taken 2 h prior to anesthesia induction decrease the preoperative fluid requirement and electrolyte deficits [1]. Thus, all patients were given a carbohydrate-containing drink and none exhibited preoperative hypotension or tachycardia; their preoperative fluid status was probably not deficient [32]. In addition, the estimated blood loss was clinically insignificant (Table 2). No patient required blood transfusion during surgery. It is thus reasonable to conclude that there is an independent association between intraoperative oliguria (with a urine output cutoff of < 0.5 mL/kg/h) and postoperative AKI.

Use of the laparoscopic approach in treatment of colorectal cancer has been shown to promote faster recovery and to reduce the length of hospital stay, volume of blood loss, and rate of complications [33–36]. It has therefore become an important component of the colorectal ERAS program [1]. Pneumoperitoneum, considered essential for adequate exposure in laparoscopic surgery, is associated with significant direct and indirect effects on renal physiology [37]. Demyttenaere et al. [38] reported that both renal function and renal blood flow tended to decrease during pneumoperitoneum. Patients in the present study underwent laparoscopic colorectal cancer surgery with the restrictive fluid therapy; our results suggested that intraoperative oliguria was associated with postoperative AKI in the context of ERAS. Further studies regarding the management of intraoperative oliguria within the colorectal ERAS program are needed.

Previous studies reported associations of perioperative AKI with short-term complications, hospital mortality, and long-term mortality [39–41]. In addition, small increases in creatinine level, which did not meet the criteria for KDIGO AKI stage 1, were associated with a twofold increase in mortality risk [42]. During follow-up, patients with intraoperative oliguria exhibited a higher rate of surgical complications than non-intraoperative oliguria patients.

This study had several limitations. First, this was a retrospective study with a relatively small sample size. However, confounding factors were adjusted between intraoperative oliguria and non-intraoperative oliguria groups by PS matching analysis. Second, our study did not evaluate the duration of intraoperative oliguria associated with postoperative AKI. Third, this study only included data from a single center. Although our ERAS protocol was based on the ERAS Society consensus guidelines, other institutions may adopt different ERAS protocols. Finally, the power to identify long-term prognosis associated with intraoperative oliguria was limited because of the relatively short follow-up period in this study.

## Conclusion

Among patients undergoing laparoscopic colorectal cancer surgery with the ERAS protocol, patients with intraoperative oliguria showed significantly increased risk of postoperative AKI, compared to those without intraoperative oliguria. Therefore, caution is needed to prevent postoperative AKI in patients with intraoperative oliguria. Further large-scale studies may be helpful to investigate appropriate management of oliguria during surgery, with consideration of long-term prognosis, within the colorectal ERAS setting.

## Supporting information

**S1 Table. ERAS protocol of Seoul St. Mary's Hospital in patients undergoing colorectal cancer surgery.**
(DOCX)

**S2 Table. Definition of acute kidney injury according to Kidney Disease Improving Global Outcomes stage.**
(DOCX)

**S1 Data.**
(XLSX)

## Acknowledgments

The authors thank Junghwa Jang and Jisun Park (Anesthesia Nursing Unit, Seoul St. Mary's Hospital, College of Medicine, The Catholic University of Korea, Seoul, Republic of Korea) for participation in our study.

## Author Contributions

**Conceptualization:** Jung-Woo Shim, Min Suk Chae.

**Data curation:** Jung-Woo Shim, Kyoung Rim Kim.

**Formal analysis:** Jung-Woo Shim, Kyoung Rim Kim, Yoonju Jung, Jaesik Park, Hyung Mook Lee, Yong-Suk Kim, Young Eun Moon, Sang Hyun Hong, Min Suk Chae.

**Investigation:** Jung-Woo Shim, Kyoung Rim Kim, Yoonju Jung, Jaesik Park, Hyung Mook Lee, Yong-Suk Kim, Young Eun Moon, Sang Hyun Hong, Min Suk Chae.

**Methodology:** Jung-Woo Shim, Min Suk Chae.

**Project administration:** Jung-Woo Shim.

**Resources:** Jung-Woo Shim.

**Software:** Jung-Woo Shim.

**Supervision:** Jung-Woo Shim, Min Suk Chae.

**Validation:** Jung-Woo Shim, Min Suk Chae.

**Visualization:** Jung-Woo Shim.

**Writing – original draft:** Jung-Woo Shim.

**Writing – review & editing:** Jung-Woo Shim, Min Suk Chae.

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
