## [Decision Letter · Decision Letter 0]

3 Mar 2020

PONE-D-20-03300

Role of intraoperative oliguria in risk stratification for postoperative acute kidney injury in patients undergoing colorectal surgery with an enhanced recovery protocol: a propensity score matching analysis

PLOS ONE

Dear DR

   Min Suk Chae 

Thank you for submitting your manuscript to PLOS ONE. After careful consideration, we feel that it has merit but does not fully meet PLOS ONE’s publication criteria as it currently stands. Therefore, we invite you to submit a revised version of the manuscript that addresses the points raised during the review process.

I would appreciate if you make careful attention in your reply to the reviewers' comments. In addition, please explain the role of vasopressors use in avoiding fluid overload and enhancing renal perfusion pressure.   

We would appreciate receiving your revised manuscript by Apr 17 2020 11:59PM. To enhance the reproducibility of your results, we recommend that if applicable you deposit your laboratory protocols in protocols.io, where a protocol can be assigned its own identifier (DOI) such that it can be cited independently in the future. For instructions see: http://journals.plos.org/plosone/s/submission-guidelines#loc-laboratory-protocols

We look forward to receiving your revised manuscript.

Kind regards,

Ehab Farag, MD FRCA FASA

Academic Editor

PLOS ONE

Journal Requirements:

2. Please ensure that you refer to Figure 1 in your text as, if accepted, production will need this reference to link the reader to the figure.

Reviewers' comments:

Reviewer's Responses to Questions

**Comments to the Author**

1. Is the manuscript technically sound, and do the data support the conclusions?

Reviewer #1: Partly

Reviewer #2: Partly

2. Has the statistical analysis been performed appropriately and rigorously? 

Reviewer #1: Yes

Reviewer #2: I Don't Know

3. Have the authors made all data underlying the findings in their manuscript fully available?

Reviewer #1: No

Reviewer #2: Yes

4. Is the manuscript presented in an intelligible fashion and written in standard English?

Reviewer #1: Yes

Reviewer #2: Yes

5. Review Comments to the Author

Reviewer #1: General Comments:

Outstanding effort and excellent manuscript, except authors fail to appreciate the potential effect of a phenylephrine infusion on renal perfusion with or without oliguria, and if this was contributory to their results and a confounding factor. This is an important flaw that must be addressed (if possible), but will require a substantial revision. Clearly, overuse of a phenylephrine infusion may mask severe hypovolemia and without data to support euvolemia was maintained or intentional hypovolemia and to what extent, it is difficult to interpret results.

Specific Comments:

Abstract: in the Patients and methods section-change "postoperative kidney functions" to "postoperative kidney function"

General Anesthesia Management Section: 1st paragraph, last sentence: either state: "The trachea was extubated" or " the endotracheal tube was removed from the trachea".

Reviewer #2: The authors aim to demonstrate that patients managed with ERAS protocol with near-zero fluid balance approach are at higher risk of postoperative AKI, however, the data does not fully support this conclusion. Specifically, ERAS protocol is not contributory to AKI in study patients as despite propensity score matching analysis we don’t know the patient’s euvolemic status at beginning of surgery. Study shows intraoperative oliguria is associated with postoperative AKI and increased morbidity but it would be wrong to interpret that it is secondary to GDFT.

ERAS recommendations:

The goal of perioperative fluid therapy is to maintain fluid homoeostasis avoiding fluid excess and organ hypoperfusion. Fluid excess leading to perioperative weight gain more than 2.5 kg should be avoided, and a perioperative near-zero fluid balance approach should be preferred. GDFT should be adopted especially in high-risk patients and in patients undergoing surgery with large intravascular fluid loss (blood loss and protein/fluid shift). Inotropes should be considered in patients with poor contractility (CI < 2.5 L/min). (1)

1. Gustafsson UO, Scott MJ, Hubner M, Nygren J, Demartines N, Francis N, et al.

Guidelines for Perioperative Care in Elective Colorectal Surgery: Enhanced Recovery After Surgery (ERAS((R))) Society Recommendations: 2018. World J Surg. 2019;43:659-695. https://doi.org/10.1007/s00268-018-4844-y PMID: 30426190

6. PLOS authors have the option to publish the peer review history of their article (what does this mean?). If published, this will include your full peer review and any attached files.

Reviewer #1: No

Reviewer #2: No

---

## [Author Response · Author response to Decision Letter 0]

21 Mar 2020

Major changes:

1. Because propensity score (PS) matching was reperformed using both the new variables in this revision and the original variables, different patients in the non-intraoperative oliguria group were PS-matched and compared to PS-matched patients in the intraoperative oliguria group. The differences between the non-intraoperative and intraoperative oliguria groups among the PS-matched patients have thus changed; all tables and figures have been revised accordingly. Although our primary outcome (group differences in the incidence of acute kidney injury) remained the same, several secondary outcomes (group differences in the incidence rates of postoperative morbidity and ileus, and the amounts of intravenous and total fluids required on postoperative day 0), have also changed. 

2. The original article listed differences in PS matching variables between the two groups in Table 1 (preoperative values) and Table 2 (intraoperative values). In this revision, all differences are shown in Table 1. The text and subsequent Tables have been modified accordingly. 

3. Of the PS matched variables, we originally included “intraoperative mean blood pressure (MBP)” in both the Methods and Results sections. We now use “average intraoperative MBP” throughout for clarity.

4. A new paragraph on the demographic characteristics of the entire study population can be found in the Results. 

5. The number of digits after each decimal point is now consistent throughout.

Journal Requirements:

Response:

The Reference style is now in accordance with PLoS ONE.

2. Please ensure that you refer to Figure 1 in your text as, if accepted, production will need this reference to link the reader to the figure.

Response:

We now refer to Figure 1.

Reviewers’ Comments:

Reviewer #1: General Comments:

Outstanding effort and excellent manuscript, except authors fail to appreciate the potential effect of a phenylephrine infusion on renal perfusion with or without oliguria, and if this was contributory to their results and a confounding factor. This is an important flaw that must be addressed (if possible), but will require a substantial revision. Clearly, overuse of a phenylephrine infusion may mask severe hypovolemia and without data to support euvolemia was maintained or intentional hypovolemia and to what extent, it is difficult to interpret results.

Response:

Thank you for this comment. If hypotension developed during surgery, patients were initially treated with 5 mg ephedrine, followed by 50 μg phenylephrine. If hypotension persisted without severe blood loss, continuous norepinephrine infusion was started at 0.03–0.10 μg/kg/min (as described in the “General anesthetic management” section).

For appropriate PS matching of potential confounding effects arising from intraoperative norepinephrine infusion and phenylephrine injection, we have added the drug doses to the intraoperative PS matched variables (as described in the “Propensity score matching variables” section and Table 1). As the intraoperative norepinephrine dose is a new PS matched variable, intraoperative norepinephrine use (a category) is removed from the PS matched variables. On reperforming the PS matching (also using variables suggested by Reviewer #2, namely the preoperative MBP and heart rate, and the average intraoperative heart rate), 125 patients from each group were selected and the AKI incidence rates were compared. As blood loss was minimal and no patient required blood transfusion during surgery (Table 2), masking of hypovolemia by intraoperatively administered vasoconstrictors is highly unlikely.

Specific Comments:

Abstract: in the Patients and methods section-change "postoperative kidney functions" to "postoperative kidney function"

Response:

We have made this change.

General Anesthesia Management Section: 1st paragraph, last sentence: either state: "The trachea was extubated" or " the endotracheal tube was removed from the trachea".

Response:

The text now reads: “The endotracheal tube was removed from the trachea”.

Reviewer #2: The authors aim to demonstrate that patients managed with ERAS protocol with near-zero fluid balance approach are at higher risk of postoperative AKI, however, the data does not fully support this conclusion. Specifically, ERAS protocol is not contributory to AKI in study patients as despite propensity score matching analysis we don’t know the patient’s euvolemic status at beginning of surgery. Study shows intraoperative oliguria is associated with postoperative AKI and increased morbidity but it would be wrong to interpret that it is secondary to GDFT.

Response:

The recent ERAS guidelines state that carbohydrate-containing drinks taken 2 h prior to anesthesia induction reduce preoperative fluid and electrolytic deficits. All patients were given a carbohydrate drink and none exhibited preoperative hypotension and/or tachycardia. Thus, their preoperative fluid status was probably not deficient. This is mentioned in the “Demographic characteristics of the entire population” section of the Results, and in the Discussion.

When reperforming PS matching of potential confounding effects caused by differences in preoperative fluid status, we added the preoperative MBP and heart rate (as described in the “Propensity score matching variables” section and Table 1). We also added the average intraoperative heart rate to the PS matched variables, to balance the intra- and preoperative vital sign variables. On reperforming the PS matching (considering also the comments of Reviewer #1), 125 patients from each group were selected and the incidence rates of AKI were compared.

---

## [Editor Report · Decision Letter 1]

25 Mar 2020

Role of intraoperative oliguria in risk stratification for postoperative acute kidney injury in patients undergoing colorectal surgery with an enhanced recovery protocol: a propensity score matching analysis

PONE-D-20-03300R1

Min Suk Chae

Dear Dr.

We are pleased to inform you that your manuscript has been judged scientifically suitable for publication and will be formally accepted for publication once it complies with all outstanding technical requirements.

With kind regards,

Ehab Farag, MD FRCA FASA

Academic Editor

PLOS ONE
---

## [Editor Report · Acceptance letter]

6 Apr 2020

PONE-D-20-03300R1 

Role of intraoperative oliguria in risk stratification for postoperative acute kidney injury in patients undergoing colorectal surgery with an enhanced recovery protocol: a propensity score matching analysis 

Dear Dr. Chae:

I am pleased to inform you that your manuscript has been deemed suitable for publication in PLOS ONE. Congratulations! Your manuscript is now with our production department. 

With kind regards,

on behalf of

Dr. Ehab Farag 

Academic Editor

PLOS ONE